# A Statistical Analysis on the Effect of Antioxidants on the Thermal-Oxidative Stability of Commercial Mass- and Emulsion-Polymerized ABS

**DOI:** 10.3390/polym11010025

**Published:** 2018-12-25

**Authors:** Rudinei Fiorio, Dagmar R. D’hooge, Kim Ragaert, Ludwig Cardon

**Affiliations:** 1Centre for Polymer and Material Technologies, Department of Materials, Textiles and Chemical Engineering, Ghent University, Technologiepark 915, 9052 Zwijnaarde, Belgium; kim.ragaert@ugent.be (K.R.); ludwig.cardon@ugent.be (L.C.); 2Laboratory for Chemical Technology, Department of Materials, Textiles and Chemical Engineering, Ghent University, Technologiepark 914, 9052 Zwijnaarde, Belgium; 3Centre for Textiles Science and Engineering, Department of Materials, Textiles and Chemical Engineering, Ghent University, Technologiepark 907, 9052 Zwijnaarde, Belgium

**Keywords:** ABS, statistical analysis, FTIR, melt processing, thermal-oxidative stability

## Abstract

In the present work, statistical analysis (16 processing conditions and 2 virgin unmodified samples) is performed to study the influence of antioxidants (AOs) during acrylonitrile-butadiene-styrene terpolymer (ABS) melt-blending (220 °C) on the degradation of the polybutadiene (PB) rich phase, the oxidation onset temperature (OOT), the oxidation peak temperature (OP), and the yellowing index (YI). Predictive equations are constructed, with a focus on three commercial AOs (two primary: Irganox 1076 and 245; and one secondary: Irgafos 168) and two commercial ABS types (mass- and emulsion-polymerized). Fourier transform infrared spectroscopy (FTIR) results indicate that the nitrile absorption peak at 2237 cm^−1^ is recommended as reference peak to identify chemical changes in the PB content. The melt processing of unmodified ABSs promotes a reduction in OOT and OP, and promotes an increase in the YI. ABS obtained by mass polymerization shows a higher thermal-oxidative stability. The addition of a primary AO increases the thermal-oxidative stability, whereas the secondary AO only increases OP. The addition of the two primary AOs has a synergetic effect resulting in higher OOT and OP values. Statistical analysis shows that OP data are influenced by all three AO types, but 0.2 m% of Irganox 1076 displays high potential in an industrial context.

## 1. Introduction

Electrical and electronic equipment (EEE) is extensively used worldwide and the production of these devices is increasing annually [1]. This results in a gradual increase in EEE waste (WEEE) generated after the end of life of these devices [1,2]. In 2016, 44.7 million metric tons of WEEE were produced worldwide and by 2021, it is expected that 52.2 million metric tons of WEEE will be created [1]. Manufacturing sustainable EEE is necessary, since the natural resources that are used to produce such devices are limited. Concerned about the increasing amount of WEEE, the European Parliament and the Council of the European Union [3] adopted the Directive 2012/19/EU, which establishes “measures to protect the environment and human health by preventing or reducing the adverse impacts of the generation and management of waste from EEE.” One of the required actions stated in this directive is associated with the recycling of WEEE, in which at least 55% of WEEE must be prepared for re-use and recycling starting in 2018.

Based on life-cycle assessment studies, the benefits of plastics recycling for the environment are evident. The recycling of plastics from WEEE results in environmental impacts about 4 times lower than those from the disposal in municipal solid waste incineration plants, and up to 10 times lower than those found in the production of virgin plastics [4]. Moreover, when considering all plastics (not only from WEEE), the use of virgin plastics instead of the recycled ones can increase the environmental impact by almost 4 times [5]. 

Among the plastics found in WEEE, styrene-based plastics such as high-impact polystyrene (HIPS) and acrylonitrile-butadiene-styrene terpolymer (ABS) are frequently present [6,7,8], and can account for more than 80 m% of the total amount [6]. Therefore, recycling HIPS and ABS is particularly important for the environment. However, it is well known that post-consumer recycled (PCR) plastics usually have different physical properties compared to those of virgin plastics. Polymers can suffer thermal-mechanical degradation during processing as well as degradation during lifetime due to many factors in the environment (heat, oxygen, light, moisture, etc.) [9]. Processing HIPS and ABS can modify physical-chemical properties due to thermal-oxidative degradation processes accelerated in the polybutadiene (PB) rich phase, causing losses in mechanical properties, as well as aesthetic modifications [10,11,12,13,14,15].

In order to improve the performance of PCR plastics, it is a common practice to blend virgin and recycled polymers or to consider suitable additives [16]. Specific additives, such as antioxidants (AOs), can be mingled with both virgin and recycled polymers to achieve the required properties [4]. There are two main types of AOs: primary AOs, which act as radical scavengers, and secondary AOs, which act as hydroperoxide scavengers [14,17]. Figure 1 shows a general scheme of the thermal-oxidative degradation of polymers without AOs (full lines) as well as in the presence of AOs (dashed lines). According to this scheme, a primary macroalkyl radical (R·) can be initially formed from a polymer chain segment (RH) due to, for example, heat (Δ), shear stress (τ), metallic impurities, or a combination of these factors, among others. Peroxide radicals (ROO·) are formed by the reaction with molecular oxygen. Then, ROO· can form hydroperoxides (ROOH) upon abstraction of hydrogen from another RH segment thereby promoting the formation of new R· species and establishing an autoxidation cycle. ROOH can decompose to alkoxy and hydroxyl radicals, which can react with RH, also producing R· species [11,14]. The addition of primary AOs (e.g., sterically hindered phenols) inhibits radical chain reactions. Additional stabilization can be obtained if secondary AOs (such as phosphites) are added, which can reduce hydroperoxides to alcohol species [18]. A synergistic effect can be obtained if different AOs are used together [17].

The influence of AOs on the stabilization of specifically PB or PB-containing polymers has been studied by several groups. De Paoli et al. [19] investigated the effects of different hindered phenols and amine stabilizers on the photooxidation of butadiene rubber, and concluded that a very effective photostabilizer is octadecyl-3-(3,5-di-tert-butyl-4-hydroxyphenyl)propionate (Irganox 1076), which is a fully sterically hindered monophenol. For thermal-oxidative degradation of a polystyrene impact-modified with PB, primary AOs such as Irganox 1076 and ethylenebis(oxyethylene)bis-(3-(5-tert-butyl-4-hydroxy-m-tolyl)-propionate) (Irganox 245), which is a partially hindered bis-phenol, inhibits degradation [14]. The influence of AOs on the thermal-oxidative stability of an ABS containing 25% polybutadiene was evaluated by Zweifel [14], and it was observed that Irganox 245 promoted higher stabilization than Irganox 1076. Földes and Lohmeijer [20] studied the performance of two commercial primary AOs (Irganox 1076 and Irganox 245) on the oxidative degradation of polybutadiene and concluded that, up to 2 m% of AO, Irganox 1076 presented better AO efficiency in oven aging (at 100 °C), whereas Irganox 245 showed higher efficiency in differential scanning calorimetry (DSC) experiments at 190 °C. Duh et al. [21] studied the oxidation of ABS (PB contents from 25 to 60 m%), using higher amounts of AO for ABS with higher PB content, and verified that the oxidation onset temperature was influenced by the AO, not by the quantity of PB. Motyakin and Schlick [22] investigated the effect of a hindered amine stabilizer (HAS, commercially known as Tinuvin 770; 1 or 2 m%) on the thermal degradation of two ABS materials (one prepared by mass polymerization containing 10 m% butadiene, and the other by emulsion polymerization containing 25 m% butadiene) and observed, at first sight, an unexpected faster degradation of polymer samples that contained a higher HAS content (2 m%). These authors also verified that the emulsion-polymerized ABS showed better thermal stability, and associated these results with the presence of a cross-linked network in the PB fraction, which is expected to promote a reduction in the free volume and oxygen diffusion [22,23]. Such PB crosslinking also occurs during aging and processing forming gels; however, the addition of AOs can improve the stability, reducing crosslinking [14,24].

According to the above studies, the oxidative stability of ABS is influenced by the type of AO as well as the polymerization method (leading to different PB amounts) in a non-trivial manner. Additional analyses are still necessary in order to evaluate the influence of the combination of different AOs, the polymerization method, and its interactions on the stability of ABS materials. The aim of the present work was to evaluate the effect of three commercially available AOs, of which two were primary and one was secondary, and two types of ABS with similar mechanical properties, either obtained by mass polymerization or emulsion polymerization, on the thermal-oxidative stability and discoloration properties. A factorial analysis of variance (ANOVA) was conducted to evaluate the effects of all factors and their interactions. To the best of the authors’ knowledge, a statistical analysis has not yet been conducted, but it is essential to further identify the dominant contributors to oxidative thermal degradation in an industrial context.

## 2. Materials and Methods

### 2.1. Materials

The materials used in this investigation were as follows: ABS Magnum 3453 Natural (Trinseo, Terneuzen, the Netherlands, obtained by mass polymerization), ABS Terluran GP-22 Natural (Ineos-Styrolution, Antwerp, Belgium, obtained by emulsion polymerization), Irganox 1076 [octadecyl-3-(3,5-di-tert-butyl-4-hydroxyphenyl) propionate] (BASF, Ludwigshafen, Germany, phenolic primary antioxidant), Irganox 245 [ethylenebis (oxyethylene)bis-(3-(5-tert-butyl-4-hydroxy-m-tolyl) propionate)] (BASF, phenolic primary antioxidant), and Irgafos 168 [tris(2,4-ditert-butylphenyl) phosphite] (BASF, phosphite secondary antioxidant). Both ABSs are general purpose injection molding grades with similar mechanical properties (Magnum 3453: density 1.05 g/cm³, melt volume rate 15 cm³/10 min (220 °C, 10 kg), tensile stress at yield (23 °C) 45 MPa, tensile modulus 2280 MPa, Charpy notched impact strength (23 °C) 20 kJ/m²; Terluran GP-22: density 1.04 g/cm³, melt volume rate 19 cm³/10 min (220 °C, 10 kg), tensile stress at yield (23 °C) 45 MPa, tensile modulus 2300 MPa, Charpy notched impact strength (23 °C) 22 kJ/m²) [25,26]. 

### 2.2. Sample Preparation and Analysis

#### 2.2.1. Simultaneous Thermal Analysis (STA)

All raw materials were simultaneously characterized by thermogravimetric analysis (TGA) and differential scanning calorimetry (DSC; STA 449 F3 Jupiter, Netzsch, Selb, Germany). Samples of 15 ± 5 mg were heated from 30 to 900 °C at a heating rate of 10 °C min^−1^, under nitrogen atmosphere (50 mL min^−1^) in a Pt-Rh pan.

#### 2.2.2. Processing

Before processing, ABS was dried at 80 °C for 4 h in a convection oven. Antioxidants (AOs) were used as received. The samples studied (see Table 1) were prepared in a micro-compounder (Minilab II, Haake Thermo Fisher Scientific, Karlsruhe, Germany) at 220 °C, 100 rpm, and the mixtures were maintained in “cycle” in the machine for 3 min. Samples were codified as follows: m% of Irganox 1076/m% of Irganox 245/m% of Irgafos 168 / type of polymerization of the ABS (M—mass polymerization; E—emulsion polymerization). For example, sample 0.2/0/0.2/M contains 0.2 m% Irganox 1076, 0.0 m% Irganox 245, 0.2 m% Irgafos 168, and the ABS processes was obtained by mass polymerization. The total mass of each processed sample was 6 g. ABS and AOs were manually added to the micro-extruder during approximately 1 minute. After processing, samples were compression molded (220 °C, 1 min, 500 kPa) into films (thickness of 250 ± 25 µm) prior to the analysis. In total, 16 experiments were performed, enabling statistical analysis and the identification of the dominant contributors.

The two virgin ABSs (unmodified; no addition of AOs and unprocessed) were also compression molded into films and their properties were evaluated for comparison. 

#### 2.2.3. Fourier Transform Infrared Spectroscopy (FTIR)

Fourier transform infrared spectroscopy (FTIR; Bruker Tensor 27, Billerica, MA, USA) analyses (for the spectra, see Appendix A) for virgin ABS and processed samples were carried out in order to evaluate chemical modifications in the PB rich phase of ABS. Samples were studied in attenuated total reflectance (ATR) mode from 4000 to 600 cm^−1^. The absorbance peaks of 1,2 butadiene at 911 cm^−1^ and 1,4 butadiene at 966 cm^−1^ were compared with the absorbance peaks of styrene (phenyl) moieties at 1603 cm^−1^ and nitrile entities at 2237 cm^−1^ using the absorption ratios *R*1 to *R*4, as defined below (Equations (1)–(4)) [15,27,28]. Values of absorbance were determined using the baseline method. With the degradation of ABS, it is expected that the nitrile and phenyl absorbances do not change and that the diene content decreases gradually, as well as its absorption peaks [13,15].(1)$$R1=\frac{{\mathrm{Absorbance}\;\mathrm{at}\;911\;\mathrm{cm}}^{-1}}{{\mathrm{Absorbance}\;\mathrm{at}\;1603\;\mathrm{cm}}^{-1}}$$
(2)$$R2=\frac{{\mathrm{Absorbance}\;\mathrm{at}\;966\;\mathrm{cm}}^{-1}}{{\mathrm{Absorbance}\;\mathrm{at}\;1603\;\mathrm{cm}}^{-1}}$$
(3)$$R3=\frac{{\mathrm{Absorbance}\;\mathrm{at}\;911\;\mathrm{cm}}^{-1}}{{\mathrm{Absorbance}\;\mathrm{at}\;2237\;\mathrm{cm}}^{-1}}$$
(4)$$R4=\frac{{\mathrm{Absorbance}\;\mathrm{at}\;966\;\mathrm{cm}}^{-1}}{{\mathrm{Absorbance}\;\mathrm{at}\;2237\;\mathrm{cm}}^{-1}}$$

#### 2.2.4. Oxidation Onset Temperature (OOT)

For the evaluation of the presence or effectiveness of AOs in a polymer, the determination of the oxidation onset temperature (OOT) is also relevant. OOT is a relative measure of the degree of oxidative stability of a material studied at certain conditions under an oxidative atmosphere; the higher the OOT value, the more stable the material [29].

The OOT of the samples was evaluated in a simultaneous thermal analyzer (STA 449 F3 Jupiter, Netzsch, Selb, Germany) based on the ASTM E2009 standard [29] (for the results, see Appendix A). All the virgin and processed samples were investigated (samples of 4 ± 0.3 mg, obtained from films of 250 ± 25 µm). The experiments were conducted under an oxygen flow of 50 mL min^−1^. A Pt-Rh pan was used, and the samples were studied from 50 to 500 °C using a heating rate of 10 °C min^−1^. The OOT was determined as the onset of the exothermic oxidation signal observed in the DSC curves. The exothermic oxidation peak temperature (OP) was also evaluated (for the results, see Appendix A). 

#### 2.2.5. Yellowing Index (YI)

The color properties of the samples were investigated with a spectrophotometer (Mercury, Datacolor, Lawrenceville, NJ, USA) using a D65 light source and 10° viewing angle. All color measurements were obtained from the compression molded films (approximately 250 ± 25 µm). The yellowing index (YI) was obtained by CIE tristimulus values X, Y, and Z according to the CIELAB color system [30]. The YI was determined according to the following:(5)$$YI=\frac{100\times (1.3013X-1.1498Z)}{Y}$$

#### 2.2.6. Statistical Analysis

A factorial design 2^4^ (four factors, two levels of each factor) was conducted to evaluate the influence of each factor (Irganox 1076, Irganox 245, Irgafos 168, and ABS type; see Table 1) and their interactions, employing analysis of variance (ANOVA) [31]. More specifically: factor ‘A’—Irganox 1076 content (levels: 0 or 0.2 m%); factor ‘B’—Irganox 245 content (levels: 0 or 0.2 m%); factor ‘C’—Irgafos 168 content (levels: 0 or 0.2 m%); and factor ‘D’—type of ABS (levels: mass- or emulsion-polymerized ABS). The values of the factors ‘A’, ‘B’, ‘C’, and ‘D’ were codified according to the level of each factor: −1 for the lower level and +1 for the higher level (for example, sample 0/0.2/0/E uses −1 for ‘A’, +1 for ‘B’, −1 for ‘C’, and +1 for ‘D’, whereas sample 0.2/0/0/M uses +1 for ‘A’, −1 for ‘B’, −1 for ‘C’, and −1 for ‘D’).

Two replications of each experiment (which reflects sources of variability both between runs and within runs [31]) were done. Since each replication, consisting of 16 experiments as shown in Table 1, was done with a time interval of approximately 20 days, replications were considered as blocks in the statistical analysis aiming to reduce nuisance factors. The order of the mixtures within each replication was randomized. The results obtained were compared using the Tukey’s test, which indicates if there is a significant difference between the average results of populations [31]; a conventional significance level (α) of 0.05 was considered.

## 3. Results and Discussion

### 3.1. Simultaneous Thermal Analysis (STA)

TGA results for the raw materials (virgin ABS by M and E; 3 AOs) are presented in Figure 2a. It is observed that the polymers show higher onsets of weight loss than those of the AOs, since AOs have lower molar masses. No significant weight loss was observed below 250 °C. Figure 2b shows the DCS results for the raw materials. It is possible to identify the rigid phase glass transition temperature (T_g_) of each ABS, close to 110 °C. The recorded melting points (T_m_) of the additives are in accordance with the data provided by the manufacturer (T_m_ of Irganox 1076: 50–55 °C (black); T_m_ of Irganox 245: 76–79 °C (red); T_m_ Irgafos 168: 183–186 °C (green)) [32,33,34]. Hence, upon processing (220 °C), all materials are in the molten state.

### 3.2. Fourier Transform Infrared Spectroscopy (FTIR)

FTIR results for the *R*1–*R*4 absorption ratios of the samples (Equation (1)–(4)) are shown in Figure 3 (black and grey symbols), including error bars. Samples from Table 1 (processed even if they do not contain AO) are organized in a descending sequence according to their experimental average *R*1–*R*4 absorption ratio value. For reference, virgin unprocessed ABS results are also shown in Figure 3 as the first two entries (blue: M; fuchsia: E), showing that the mass-polymerized ABS has likely a lower PB content despite the mechanical properties of the ABS materials studied being similar according to the manufacturers [25,26]. Gesner studied ABSs with different PB content, and observed that a higher *R*2 indicates a higher PB content [35]. In agreement with this statement one can observe that the processed samples containing emulsion-polymerized ABS tend to present higher *R*1–*R*4 (black symbols) absorption ratios than the samples based on ABS obtained by mass polymerization (grey symbols).

The ANOVA results for both *R*1 and *R*2 absorption ratios indicate that the only statistically significant factor is the factor ‘D’ (type of ABS; *p*-values <0.0062), explaining the alignments of the predicted values (red symbols in Figure 3a,b—Equations (6) and (7)). The replications (blocks) in the analysis of variance of *R*1 and *R*2 showed a *p*-value of 0.0148 and 0.0370, respectively, indicating nuisance factors influenced the results in each block. (6)R1=1.104+0.154D
(7)R2=1.722+0.139D

However, the ANOVA results for the *R*3 absorption ratio indicate that several factors and interactions are statistically significant: ‘B’ (Irganox 245 content; *p*-value of 0.0401), ‘D’ (type of ABS; *p*-value <0.00001), ‘AC’ (Irganox 1076 and Irgafos 168; *p*-value of 0.04875), and ‘BD’ (Irganox 245 and type of ABS; *p*-value of 0.04346). The ANOVA results for the *R*4 ratio show that the significant factors and interactions are as follows: ‘B’ (Irganox 245 content; *p*-value of 0.01434), ‘D’ (type of ABS; *p*-value <0.00001), ‘AD’ (Irganox 1076 and type of ABS; *p*-value of 0.01212), and ‘BD’ (Irganox 245 and type of ABS; *p*-value of 0.0478). The replications (blocks) showed a *p*-value of approximately 0.001 for both *R*3 and *R*4. The predicted values presented in Figure 3c,d (red symbols) were obtained from Equations (8) and (9):(8)R3=1.622+0.015A+0.041B−0.008C+0.039AC+0.296D+0.041BD,(9)R4=2.522+0.061A+0.086B+0.315D+0.088AD+0.067BD.

Hence, from Figure 3c,d it follows that *R*3 and *R*4 absorption ratios (reference peak: absorbance of nitrile groups at 2237 cm^−1^) are more sensitive to modifications in 1,2 butadiene and 1,4 butadiene absorptions peaks than *R*1 and *R*2 (reference peak: absorbance of styrene moieties at 1603 cm^−1^). Previous studies indicated an increase in FTIR absorption spectra at approximately 1600 cm^−1^ for photodegraded ABS, and it was associated (see Figure 1) with the formation of carbonyl groups [27,36]. These carbonyl moieties can be related to the oxidative degradation of the PB fraction [11], and also to the presence of primary AOs, since both primary AOs investigated in this work exhibit C=O groups. The simultaneous change of absorptions around 1600 cm^−1^ therefore obscured the modifications of PB phase for *R*1 and *R*2. From the Tukey’s multiple comparison test (not shown here), and considering the experimental average values of *R*1 and *R*2 absorption ratios, no difference was observed among the samples. This indicates that the use of the absorption peak at 1603 cm^−1^ as reference peak is insensitive to indicate changes in the PB fraction of the ABSs due to thermal-oxidative degradation, since the FTIR absorption at 1600 cm^−1^ also changes [27,36].

Figure 4 shows the Tukey’s test results for *R*3 and *R*4 absorption ratios, aiming to clarify which samples showed undeniable differences between the average results obtained. One can observe that there is a clear statistical difference among samples according to the type of ABS used. No difference was observed among all samples from mass-polymerized ABS (see Figure 3), implying that the change of the PB phase is limited. For emulsion-polymerized ABS, only the sample 0/0/0.2/E showed statistically different values, presenting a lower *R*3 and *R*4 value in comparison with sample 0.2/0.2/0.2/E and also a lower *R*3 value related to the sample 0/0.2/0/E. If one solely relates degradation with the changes in the PB phase, one expects at first sight that the sample 0.2/0.2/0.2/E would show higher absorption ratios and consequently a higher thermal-oxidative stability. However, the addition of AOs did not cause a statistically significant difference in the absorption ratios in comparison to the sample 0/0/0/E. This could imply that degradation is not dominant in the PB phase (or very limited). Nevertheless, these insights could still alter if more repeat experiments are conducted, other analysis properties are considered (see discussion below), a lower AOs content is used (<0.2 m%), the melt processing time is higher (>3 min), or the melt processing temperature is higher (>220 °C). The variation of the preceding process parameters is out of the scope of the present work.

### 3.3. Oxidation Onset Temperature (OOT)

OOT results are presented in Figure 5a, following the same color coding as in the Figure 3 ranking again as the processed samples in Table 1, from high to lower stability, and again adding the virgin unprocessed data. A different ranking from the results in Figure 3 occurs, highlighting that a sole focus on the PB content is insufficient. The ANOVA results show that the factors ‘A’ (Irganox 1076), ‘B’ (Irganox 245), and ‘D’ (type of ABS) as well as the interaction ‘AB’ (between Irganox 1076 and Irganox 245) significantly affected the OOT of ABS; these factors and the interaction exhibit *p*-values lower than 0.01. Irgafos 168 (factor ‘C’) shows a *p*-value of 0.077 and was not considered significant for the OOT results. From the ANOVA results of OOT data, Equation 10 is obtained, presenting an acceptable correlation coefficient (R) of 0.853, as is also reflected by the relatively good agreement between experimental and predicted results observed in Figure 5a.(10)$$\mathrm{OOT}\;\left({}^{{}^{\circ}}C\right)=189.19+7.551A+4.915B-3.578\mathrm{AB}-3.781D$$

From Figure 5a, one can observe that the processing of both virgin (unmodified) ABS materials (last two entries) causes a very strong decrease in OOT in comparison with the virgin unprocessed ABS materials (first two entries). This reduction in OOT is likely related to the consumption of AOs added by the polymer manufacturers during the original manufacturing and at least partly due to the degradation of the PB phase [14]. The addition of primary AOs promotes an increase in polymer stability of the studied samples. The stabilization effect of the primary AOs was expected, as it was also verified in previous studies [14,19,20]. The OOT values observed for samples containing at least one of the AOs studied are close to those observed for virgin (unprocessed) ABS, further indicating that the individual amount of 0.2 m% can be critical. Equation (10) predicts that the highest OOT values are found in samples containing both Irganox 1076 (factor ‘A’) and Irganox 245 (factor ‘B’) (samples 0.2/0.2/0/M, 0.2/0.2/0.2/M, 0.2/0.2/0/E, and 0.2/0.2/0.2/E). Samples containing only the secondary AO (Irgafos 168; factor ‘C’, 0/0/0.2/M and 0/0/0.2/E) show lower OOT values than those observed for the samples containing one (or both) primary AOs. Given that the OOT is obtained at the beginning of the oxidation process, it is expected that primary AOs (radical scavengers) affect the OOT values more than the secondary AOs do, as the latter act as hydroperoxide scavengers. Secondary AOs do not operate in the first steps of the thermal-oxidation process (where OOT is identified) but can interfere in subsequent degradation steps (see Figure 1), which explains Irgafos 168’s lack of influence on OOT, according to the ANOVA results.

In addition, the type of ABS (factor ‘D’) significantly influences the OOT results. The ABS obtained by mass polymerization presents a higher thermal-oxidative stability than the emulsion-polymerized ABS. This higher stability for the mass-polymerized ABS may be related to the expected lower PB content of such samples (see the FTIR data in Figure 3), and it is likely also related to the differences in the morphology of the PB phase of studied ABS types. Giaconi et al. [37] observed that the morphology of the PB phase found for mass and emulsion-polymerized ABS materials is different. A mass polymerization-based ABS shows a ”salami” morphology, in which the butadiene disperse phase shows inclusions of styrene-acrylonitrile copolymers (SANs) inside the dispersed PB particles, whereas an emulsion-polymerized ABS shows bulk rubber particles, with almost no SAN inclusions in the butadiene phase [37]. The differences in morphology can interfere in the thermal-oxidative resistance of the studied ABSs, since the inclusions of SAN phase in the butadiene particles can inhibit oxidation of the rubber phase, increasing the OOT values. Note that the present results are at first sight contradictory to the work of Motyakin and Schlick [22], in which the emulsion-polymerized ABS presented higher thermal stability, even containing a higher PB content than the mass-polymerized ABS. This diverging behavior can be related to the presence of a hindered amine stabilizer (HAS) in that work, the thermal aging of the samples at only 120 °C, and the different stabilizer amount (1 or 2 m%). Additionally, those authors did not investigate the unmodified ABS. Figure 5b shows the Tukey’s test results for the OOT experiments. Almost all samples show statistically equivalent OOT values, despite the fact that an increasing trend (from right to left) can be observed overall for the red points (predicted values) in Figure 5a, highlighting the difference between measurements and detailed statistical analysis. Importantly, an exception is the increase in OOT for samples containing Irganox 1076 (left region of Figure 5a).

Figure 6 shows the OP results for the samples investigated—with the same ranking as in Figure 5a, from high to low stability (left to right)—and the respective Tukey’s test results. The ANOVA results shows that the four main factors ‘A’ (Irganox 1076), ‘B’ (Irganox 245), ‘C’ (Irgafos 168), and ‘D’ (type of ABS) significantly affect the oxidation peak (*p*-values lower than 0.005). Moreover, the interaction ‘AB’ (between Irganox 1076 and Irganox 245; *p*-value of 0.00226) and the interaction ‘ABD’ (between Irganox 1076, Irganox 245, and type of ABS; *p*-value of 0.0237) also influence OP. From these results, Equation (11) is obtained, leading to a very good prediction (R = 0.966), further highlighting the need of multiple analyses.(11)$$\mathrm{OP}\;\left({}^{{}^{\circ}}C\right)=206.71+5.647A+5.109B-1.628\mathrm{AB}+1.478C-4.997D+1.116\mathrm{ABD}$$

From Equation (11), it is possible to predict that the samples with the highest OP values for each ABS are those containing the three AOs simultaneously, implying that an interpretation solely on statistical analysis of absorption data (see Figure 3 and Figure 4) or OOT data (see Figure 5) is not recommended. Similar to the effect observed for the OOT values, Figure 6a shows that the processing of both unmodified ABS types causes a significant decrease in OP due to, for example, PB-phase degradation during processing and that the addition of primary AOs, as well as the type of ABS, influences the OP values. However, the secondary antioxidant (Irgafos 168, factor ‘C’) shows a significant influence on the OP, a phenomenon not observed in OOT for this AO. The influence of the secondary AO is observed in the oxidation peak values but not in OOT since the OP occurs at the maximum stage of the oxidation process, and it is expected that at this point significant amounts of hydroperoxides are being formed (see Figure 1), which allows the secondary AO (a hydroperoxide scavenger) to be effective. Furthermore, the mass-polymerized ABS showed higher OP values than those found for the emulsion-polymerized ABS, considering the same AO composition, indicating that the amount of PB and the morphology of PB phase interfere in the thermal-oxidative stability of ABS, as well as for the OP. From Figure 6b (Tukey’s test), it follows that samples containing at least one primary AO show higher OP values, whereas the addition of the secondary AO alone results in lower OP values. Hence, despite the fact that the secondary AO significantly affects the OP values, the primary AOs have a more pronounced effect on OP.

### 3.4. Yellowing Index (YI)

The yellowing index (YI) results are shown in Figure 7, ranking the data in Table 1 from low to higher values (right to left) and complementing this data set with the data for virgin ABS. The ANOVA results for the yellowing index (YI) show that the type of ABS (factor ‘D’; *p*-value lower than 0.0002) and the interaction between Irganox 1076 and Irganox 245 (‘AB’; *p*-value lower than 0.0271) are significant. The replications (blocks) for ANOVA of the YI show a *p*-value <0.00001, indicating a strong difference concerning the replications. From these results, and after removing the effect between replications (blocks), Equation (12) is obtained. A lower correlation coefficient (R = 0.744) was obtained for this equation, mainly related to nuisance factors (e.g., possible differences in the cleanliness of the micro-extruder).(12)YI=11.16−0.593A+5.551B−1.211AB+2.433D

From Figure 7 and Equation (12), one can observe that the processing of both ABS types causes an increase in the YI. The type of ABS largely influences the YI and the virgin ABS obtained by mass polymerization presents a lower YI than the virgin emulsion-polymerized ABS. This difference in the YI is again related to the higher amount of PB phase of the ABS based on emulsion polymerization. It should be noted that Zweifel [14] reported that styrene-acrylonitrile copolymers (SAN) become discolored above 220 °C even in the absence of oxygen due to reactions of the acrylonitrile comonomer, and the combination of Irganox 1076 with Irgafos 168 suppresses discoloration during processing. However, the data analysis in the present work (see Figure 3 and Figure 7) indicates that the PB significantly contributes to the increase in the ABS YI. In addition, the multiple comparison Tukey’s test for the YI results (not shown here) indicate no significant difference among all processed samples. This is probably related to a variation in the cleanliness of the equipment before each processing, which generated an additional variance for the YI results.

### 3.5. Summary of the ANOVA

Table 2 summarizes the results for the different degradation properties obtained in the present work according to ANOVA (for the results, see Appendix A). For clarity, the reader is reminded that (−1) and (+1) represent the level of each factor which results in the higher predicted value for each property. By jointly investigating these properties, the relevance of more individual and interaction parameters is evident. It is observed that the type of ABS affects all properties and is thus very significant for the interpretation of the degradation data. The emulsion-based ABS (‘D’ = +1) provides higher values for the *R*1–*R*4 absorption ratios (more PB phase), as well as higher YI. The mass-polymerized ABS shows higher OOT and OP values after melt processing, indicating better thermal-oxidative stability. The addition of both Irganox 1076 and Irganox 245 results in higher OOT and OP values, and there is a synergistic effect with the use of these primary AOs, as can be deduced from the significant interaction AB. However, this effect can be related to the higher amount of primary AOs in the polymer when using Irganox 1076 and Irganox 245 simultaneously. The addition of 0.2 m% of Irganox 1076 is however better than adding 0.2 m% of Irganox 245. The secondary AO has a significant effect on OP, increasing the thermal-oxidative stability. Only the OP displays a relevance for all AOs and several interactions, as witnessed by the significance of the interaction ‘AB’ and ‘ABD’.

## 4. Conclusions

Processing of unmodified emulsion- and mass-polymerized ABS (220 °C) significantly degrades the polybutadiene (PB) phase, contributing to the reduction of the oxidation onset temperature (OOT) and the oxidation peak temperature (OP), as well as to the increase in the yellowing index (YI). Therefore, antioxidants (AOs) should be added in sufficiently high amounts during the original processing to reduce thermal-oxidative degradation. It can be postulated that this results in post-consumer recycled plastics—such as those originating from waste electrical and electronic equipment (WEEE)—with better physical-chemical properties.

The relative PB phase amount variation for processing at 220 °C can be assessed by dedicated Fourier transform infrared spectroscopy (FTIR). The styrene (phenyl) absorption peak at 1603 cm^−1^ changes simultaneously with the thermal-oxidative degradation of the PB rich phase, whereas the nitrile absorption peak at 2237 cm^−1^ can be seen as insensitive to this degradation. Therefore, the nitrile absorption peak is recommended as reference peak in order to evaluate chemical changes in the PB content.

Due to a lower PB content, the ABS samples obtained by mass polymerization display higher thermal-oxidative stability than those samples obtained with emulsion-polymerized ABS, despite the fact that both virgin ABS types show similar mechanical properties. Hence, the use of an ABS polymerized by a mass process in the manufacturing of EEE can generate WEEE ABS with lower changes in physical-chemical properties than WEEE obtained from emulsion-polymerized ABS.

The addition of the two primary AOs studied (Irganox 1076 and Irganox 245) enhances the thermal-oxidative stability, increasing both OOT and OP, as well as causing a reduction in the YI. A synergetic effect is observed with the combination of the two primary AOs, resulting in higher OOT and OP values. The addition of the secondary AO increases OP, further increasing thermal-oxidative stability. This OP is influenced by all AO types and several interactions, as can be deduced from the ANOVA predictive equation. Further statistical analysis of the current data set however suggests that the use of 0.2 m% of primary AO, in particular Irganox 1076, displays a high potential, which is economically attractive.

## Figures and Tables

**Figure 1 polymers-11-00025-f001:**
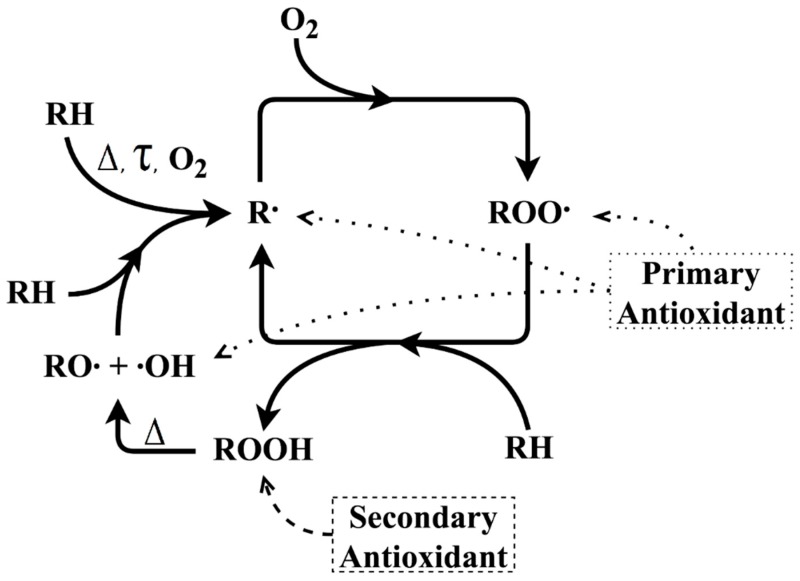
General scheme of thermal-oxidative degradation of polymer (depicted for segment RH) and its inhibition by primary and secondary antioxidants (adapted from [14]).

**Figure 2 polymers-11-00025-f002:**
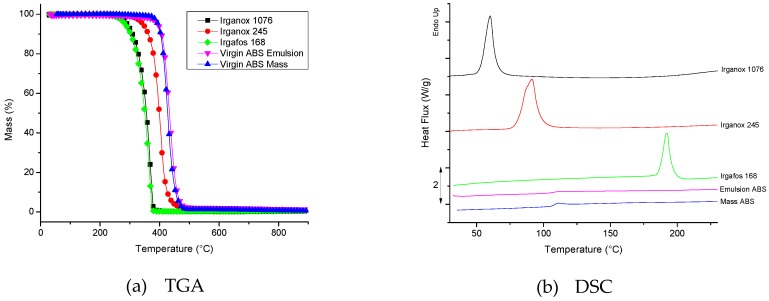
(**a**) Thermogravimetric (TGA) results; and (**b**) Differential scanning calorimetry (DSC) results for the raw materials studied.

**Figure 3 polymers-11-00025-f003:**
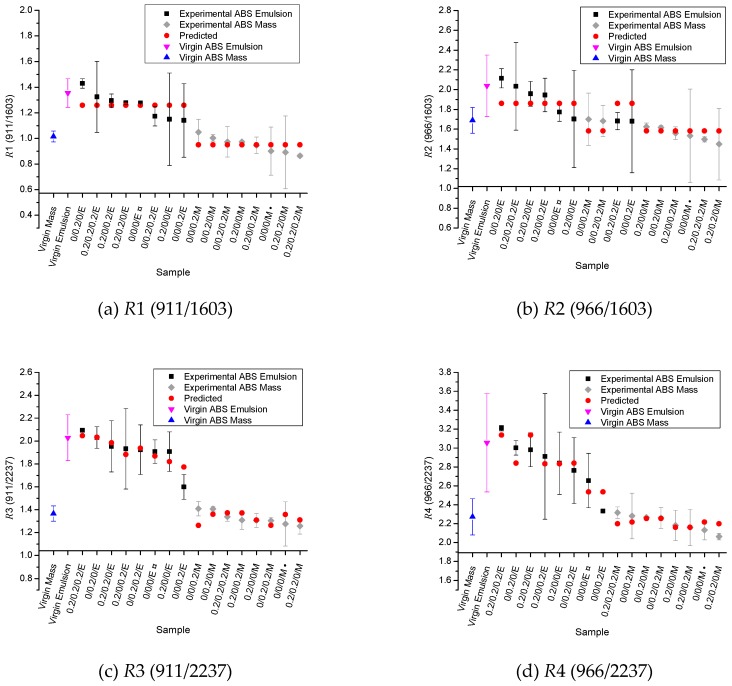
Measured (black (emulsion-polymerized ABS) and grey (mass-polymerized ABS)) and predicted (red) FTIR absorption ratios for samples presented in Table 1, including additionally the virgin unprocessed ones: blue: mass, and fuchsia: emulsion-polymerized ABS; (**a**) *R*1 (911/1603); (**b**) *R*2 (966/1603); (**c**) *R*3 (911/2237); (**d**) *R*4 (966/2237); Equations (1)–(4) for definitions; predictions based on Equations (6)–(9); *R*3 and *R*4 data are most suited for statistical analysis; entries from Table 1 ranked from high to lower absorption ratios (left to right); data highlight a lower polybutadiene content for mass-polymerized ABS, consistent with the lower blue (virgin unprocessed) point.

**Figure 4 polymers-11-00025-f004:**
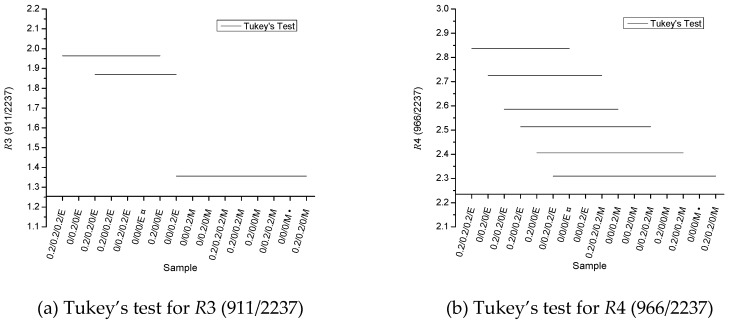
Tukey’s test for the two most sensitive FTIR ratios from Figure 3: (**a**) *R*3 (911/2237); (**b**) *R*4 (966/2237).

**Figure 5 polymers-11-00025-f005:**
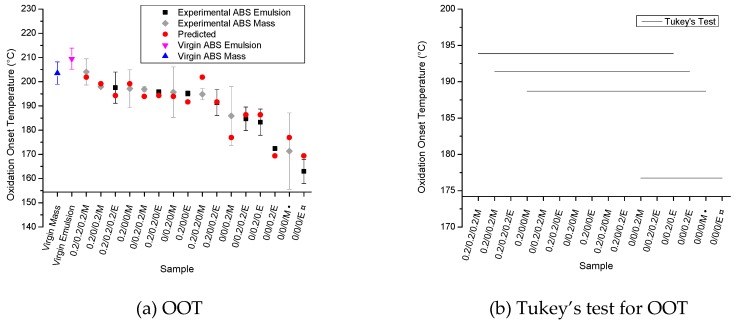
Oxidation onset temperatures (OOTs) for the samples studied, with a ranking from high to low stability (left to right); (**a**) OOT results (color coding as in Figure 3; Equation (10) for prediction); (**b**) Tukey’s test for OOT results; data indicate an undeniable higher stability for mass-polymerized ABS.

**Figure 6 polymers-11-00025-f006:**
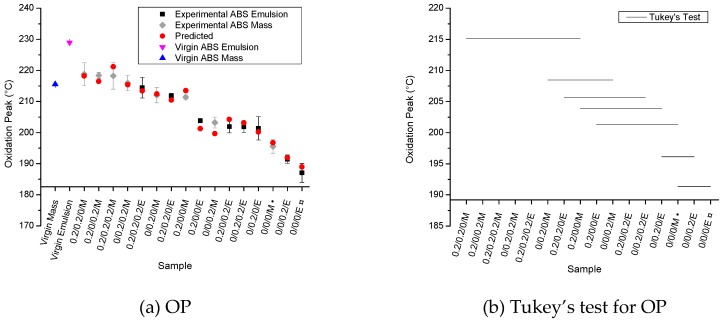
Oxidation peak (OP) for the samples studied; (**a**) OP results (color coding again as in Figure 3, ranking from high to low stability (left to right)); (**b**) Tukey’s test; consistent with Figure 5, the data highlight a high stability for mass-polymerized ABS.

**Figure 7 polymers-11-00025-f007:**
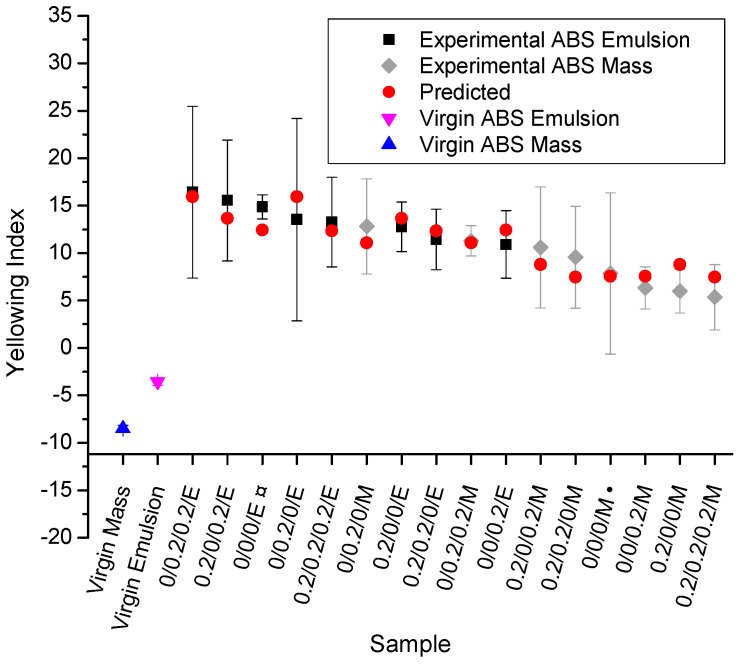
Yellowing index (YI) for the samples studied in Table 1 (ranking from low to high, right to left) and the two unprocessed virgin ones (outer left); again, color coding is as in Figure 3; consistent with the data in Figure 5 and Figure 6, more yellowing takes place for emulsion-polymerized ABS.

**Table 1 polymers-11-00025-t001:** Composition of the processed acrylonitrile-butadiene-styrene terpolymer/antioxidant (ABS/AO) combinations; M: mass; E: emulsion; numbers are m% in the order of the columns; a sufficient variation of the process variables is considered to allow for statistical analysis of the dominant factors.

Experiment	Sample Code	Irganox 1076 (m%)	Irganox 245 (m%)	Irgafos 168 (m%)	Type of ABS
1	0/0/0/M	0.0	0.0	0.0	Mass
2	0.2/0/0/M	0.2	0.0	0.0	Mass
3	0/0.2/0/M	0.0	0.2	0.0	Mass
4	0.2/0.2/0/M	0.2	0.2	0.0	Mass
5	0/0/0.2/M	0.0	0.0	0.2	Mass
6	0.2/0/0.2/M	0.2	0.0	0.2	Mass
7	0/0.2/0.2/M	0.0	0.2	0.2	Mass
8	0.2/0.2/0.2/M	0.2	0.2	0.2	Mass
9	0/0/0/E	0.0	0.0	0.0	Emulsion
10	0.2/0/0/E	0.2	0.0	0.0	Emulsion
11	0/0.2/0/E	0.0	0.2	0.0	Emulsion
12	0.2/0.2/0/E	0.2	0.2	0.0	Emulsion
13	0/0/0.2/E	0.0	0.0	0.2	Emulsion
14	0.2/0/0.2/E	0.2	0.0	0.2	Emulsion
15	0/0.2/0.2/E	0.0	0.2	0.2	Emulsion
16	0.2/0.2/0.2/E	0.2	0.2	0.2	Emulsion

**Table 2 polymers-11-00025-t002:** Summary of the ANOVA results for the studied degradation properties, based on the experimental data in Figure 3, Figure 4, Figure 5, Figure 6 and Figure 7 and Equations (1)–(12); S: significant; I: insignificant; (−1) or (+1): level of each factor which results in the higher predicted value for each property; M/E: −1/+1.

Property	Main Factor (Code)				Significant Interaction(s)
	Irganox 1076^a^ (‘A’)	Irganox 245^a^ (‘B’)	Irgafos 168^b^ (‘C’)	Type of ABS (‘D’)	
*R*1	I	I	I	S (+1)	-
*R*2	I	I	I	S (+1)	-
*R*3	I	S (+1)	I	S (+1)	AC; BD
*R*4	I	S (+1)	I	S (+1)	AD; BD
OOT	S (+1)	S (+1)	I	S (−1)	AB
OP	S (+1)	S (+1)	S (+1)	S (−1)	AB; ABD
YI	I	I	I	S (+1)	AB

^a^ Primary antioxidant (AO); ^b^ Secondary AO.

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
