# Peer review of "A Statistical Analysis on the Effect of Antioxidants on the Thermal-Oxidative Stability of Commercial Mass- and Emulsion-Polymerized ABS"

_polymers, 2018, doi:10.3390/polym11010025_

Round 1
Reviewer 1 Report
My general feeling is that the manuscript has the required quality to be considered for publication in Polymers. Nevertheless, before possible publication, authors should individually address the following comments/suggestions/questions:
Abstract
- First sentence: “Melt processing of acrylonitrile-butadiene-styrene terpolymer (ABS) causes modification of the physicochemical properties.” – Seems too generic and out-of-place. Consider removing it or moving to a different part of the Abstract.
- “Fourier transform infrared spectroscopy with as reference the absorbance for the nitrile entities at 2237 cm-1 24 is recommended to identify changes in the PB content.” – Rewrite
- “The 27 addition of a primary AO increases the thermal-oxidative stability with a synergistic effect for the 28 combination, …” - Rewrite
- “Oxidation peak (OP)” – Does it refer to the temperature corresponding to this peak?
- “. Statistical analysis shows that OP data are influenced by all three AO types but as such 0.2 m% of Irganox 1076 displays high potential in 30 an industrial context.” – Revise (poor English); “0.2 wt%” instead of “0.2 m%” (and throughout the manuscript). Why “0.2 wt%”? Is this a typical value found in similar systems?
Introduction
- “…as well as degradation during lifetime due to many factors in the environment (heat, oxygen, light, moisture etc.) [9].” – “…as well as degradation during lifetime due to many factors in the environment (heat, oxygen, light, moisture, etc.) [9].
- “…and observed an at first sight an unexpected faster degradation of polymer samples that contained a higher HAS content” – Replace by “…and observed at first sight an unexpected faster degradation of polymer samples that contained a higher HAS content”
- “Such PB crosslinking also occurs during aging and processing, forming gels but the addition of AOs can improve the stability” - Rewrite
Materials and methods
- Authors should give more details about the two types of ABS (at least their density and especially their MFI or MVR)
- As you know, TGA-DSC equipments are not thought to run precise DSC measurements, but instead thought to analyze endothermal or exothermal transitions of materials such as metal hydroxides or other inorganic materials found at high temperatures. As a consequence, take care when using said values to run endothermal/exothermal transition analyses of ABS. Why only in nitrogen and not in air? Have you considered the possibility of running TGA-DSC tests using air?
- Why “220 ºC”? Is this processing temperature the result of prior optimization or was it randomly selected?
- “ABS and AOs where manually added to the micro-extruder” – How? Little by little at a specific feeding rate or at once?
- It seems that the films where obtained with considerable thickness variation. Was this a consequence of non-proper compression-moulding variables or differences in material viscosity prior to compression-moulding?
- Why +1 for “E” and -1 for “M” (type of ABS)?
Results and discussion
- “Figure 2(b) shows the DCS results DSC for the raw materials.” – Replace by “Figure 2(b) shows the DCS results for the raw materials.”
- Consider enhancing the size of Figures 2(a) and 2(b) (too small to see possible differences)
- “…showing that the mass polymerization ABS has likely a lower PB content despite the mechanical properties of the ABSs materials studied being similar according to the manufacturers [25,26].” – Can you quantify it?
- Some of the error bars presented in Figure 3 (also in Figure 7) are very high. Do you have any explanation for such error bars? Such error bars make difficult for any further proper analyses. I am not sure if with these error bars enable to validate the following conclusions.
- “…further indicating that the individual amount of 0.2 m% can be critical.” – Critical meaning “minimum required concentration”? Explain.
- You mention “differences in the morphology of the PB phase of studied ABS types” – Can you demonstrate this statement?
- “was obtained for this equation, mainly related to nuisance factors (e.g. possible differences in the cleanliness of the micro-extruder)” – Does this fact take out the validity and usefulness of the presented results?
Conclusions
- As it is stated that differences in oxidation are basically related to the amount of PB phase and its morphology, authors should have presented the calculus of the amount of PB in each type of ABS and demonstrate possible differences in PB’s phase morphology.
Author Response
Dear Reviewer,
The authors are very grateful to the reviewers for their careful and meticulous reading of the manuscript. The reviews are detailed and helpful to finalize the manuscript. The authors would like to kindly acknowledge them.
Attached are the answers to the comments of all reviewers.
Best regards,
Rudinei Fiorio

Reviewer 2 Report
The effect of three antioxidants on two different types of ABS has been evaluated through FTIR, oxidation onset temperature, oxidation peak temperature and yellowing index. Statistical analyses showed differences according to the agent and the employed polymers. In general, the work seems well performed, although probably its length could be easily shortened.
Four minor points should be clarified before acceptance:
a) The two selected ABS samples should be better described in therms of physical properties.
b) Selected processing conditions should also be better justified.
c) Differences concerning morphology of the polybutadiene phase as well as its content in each blend should be given,
d) Error bars for R1, R2, R3 and R4 ratios are in some cases enormous (see for example R4 for the 0.2/0/0/E sample). Some explanations should be given in those cases. Is the statistical analysis correct considering such errors?
Author Response

(The authors gave the same response as above.)

Reviewer 3 Report
polymers-405709
The authors present in their work entitled ‘A statistical analysis on the effect of antioxidants on the thermal-oxidative stability of commercial mass- and emulsion-polymerization ABS’ a systematic study comprising statistical analyses on the effects of various antioxidants in the recycling process of ABS.
The idea and the approach taken are valid; the experimental work underlying the statistical analyses is described in sufficient detail. A statistical analysis is, as such, eventually yet less commonly used for the problem under study, but quite valid. The authors correctly state that the interplay of various factors contributing to the effect of antioxidants in polymerisation processes is complex, and eventually difficult to delineate due to synergistic effects, etc. A statistical analysis taking into account the multivariability is thus an interesting approach worth being disseminated as such. The question is whether the amount of experimental data, and especially their quality is really sufficient enough to actually allow the statistical approach used to deliver as many insights as eventually possible. The weak point of the paper is somewhat the fact that the statistical analyses has a hard time to actually reveal something hidden that would be impossible to reveal using a series of chemical analysis.
However, given the general and principal potential of the chosen approach, the paper is worth being published in an open access journal concerned with polymer processing.
Some aspects should be looked at during a revision:
· The experimental part should be liberated from any explanations not necessary for the technical work. These should be rigorously moved into the discussion.
· The statistical methods used should be briefly explained and rationalised, given that the topic is (yet) not often encountered in a chemistry journal.
· For the same reason, the supporting information should be extended: the authors should present the formulas used within the exploited statistical methods and roughly indicate the emergence of the factors presented in the main text in the various formulas.
Author Response

(The authors gave the same response as above.)
